# Barriers to Applying Last-Mile Logistics in the Egyptian Market: An Extension of the Technology Acceptance Model

Mohamed Amr Sultan [1,2,*], Tomaž Kramberger [2], Mahmoud Barakat [1] and Ahmed Hussein Ali [1]

1 College of International Transport and Logistics, Arab Academy for Science, Technology and Maritime Transport, Alexandria 1209, Egypt; mahmoud.barakat@aast.edu (M.B.); ahmed.husseincitl@aast.edu (A.H.A.)
2 Faculty of Logistics, University of Maribor, 3000 Celje, Slovenia; tomaz.kramberger@um.si
* Correspondence: amrsultan@aast.edu; Tel.: +20-1001410033

**Abstract:** Drawing on the Technology Acceptance Model (TAM), this research aims to investigate the impact of social, environmental, and technological barriers on adopting the last-mile logistics application. This research used a self-administrative questionnaire to collect 1060 respondents from the Egyptian market and analysed it using partial least square structural equation modeling. The findings revealed that some elements could obstruct the implementation of last-mile delivery technologies, namely complexity, collaboration efforts between users and application developers and the impact of technical knowledge and expertise on the potentially involved users. The sharing economy helps organisations reduce contaminants, emissions and carbon footprints, and last-mile logistics is one of the tools of the sharing economy that can enhance the productivity and competitiveness of logistics and boost consumer fulfillment. This research will help enhance organisations' performance in Egypt as a developing country and push towards applying environmental sustainability practices, as it introduces a tool to enhance customer satisfaction and reduce emissions by illustrating how last-mile logistics can be implemented. This is particularly important as last-mile logistics face some implementation barriers, especially in developing countries. In addition, it will help in extending the theory through conceptualising its abstract ideas with the research variables and applying it in a different context.

**Keywords:** technology acceptance model; technology implementation barrier; last-mile delivery; sustainability; sharing economy; structural equation modelling





## 1. Introduction

The speedy progress of the sharing economy and its extraordinary influences on numerous aspects of today's social, economic scheme and industrial life has aroused increasing public significance in the last few years among different stakeholders [1]. This speedy growth of the sharing economy in the past era is significantly associated with social, industrial and economic environments in the quest for the better value distribution of the supply chain, a decrease in environmental impacts, the development of technology and, eventually, users' changing approaches concerning product possession and the need for social assembly [2]. The main objective of the sharing economy is to maximise results while consuming as few resources and services as possible for business operations. It includes many initiatives [3]; these initiatives can be carried out individually through a platform to conduct exchanges, rentals, donations or sales. In addition, central access to idle resources makes them usable [4,5]. The sharing economy refers to a broad range of a high proportion of attention has been focused on their capability to elevate substantial financing quantities and correspond to their extraordinary worldwide development. Depending on a virtual platform to expediently link users and providers, these businesses have achieved immediate international acceptance, which has developed resoundingly with regional competitors and managers. It is an online platform used to share the needs of a community

to make perfect deals, increase consumer economic practices and increase their attention to sustainable development goals [6,7]. A significant portion of the sharing economy creates physical goods flows in cities, boosting deliveries there and diversifying the types of logistics resources that are mobilised [8]. As a result, recent research has begun to outline a new, promising area of study at the intersection of the sharing economy and urban logistics through adopting the Technology Acceptance Model (TAM) [5,9]. Davis suggested the Technology Acceptance Model (TAM) in 1989, which aims to evaluate or understand the practices and conduct of information technology consumers. The technology acceptance model can generally explain how outside factors impact the consumer's inner "attitude", "belief" and "behavioural intention" [10]. The research proposed factors of perceived usefulness (PU) and perceived ease of use (PEU) to justify and guess an individual's acceptance of technology and to evaluate the factors influencing an individual's acceptance of new information. This supports that perceived usefulness and ease of use will impact attitudes concerning technology usage, thus impacting the specific behaviour [11,12]. As explained, this research will use the TAM and extend it to our research on how individuals will accept the concept of the sharing economy in last-mile delivery through our proposed model. The MAY-D system is one of those platforms used to exchange the transport services of passengers or packages. The users can provide the services or request them according to their needs. This system engages citizens in last-mile delivery to improve urban logistics and achieve sustainability goals [13,14]. Last-mile delivery (LM) is an expression used in supply chain management (SCM) and transportation planning to portray the movement of people and goods from a transportation hub to a domestic destination [15]. "Last-mile delivery" is a logistics perception that includes tailored processes to guarantee that final delivery is comfortable for shoppers and efficient [16,17]. Last-mile delivery can be defined as the transportation of goods from a retailer to the end consumer, whilst last-mile logistics describes the processes and systems that enable that delivery [18].

Like any human curiosity, online trade puts pressure on the environment, specifically the urban environment, where most inhabitants live. Logistic facilities, including those associated with last-mile deliveries of electronic shopping, are significant contributors to heightened emissions; an increase of one-third is anticipated. In addition, metropolitan overcrowding associated with last-mile deliveries is projected to rise by 21% by 2030 [19]. This gives the impression that our cognitive capacities, which were beneficial in the past for human beings to continue to exist, such as reducing the predilection for short-term momentary incentives over greater but more immediate ones, might lead to catastrophic outcomes. The compulsion for accessibility, one click-spending and a special rate of returns are ruthlessly unfavorable to the environment. This gives the impression that people regularly do not worry enough about the long-term consequences of their actions, especially concerning their environmental effects. However, several papers recommend that electronic shopping may be more eco-friendly than traditional shopping [20,21]; especially in non-food shopping, home delivery is favourable in terms of $CO_2$ production [22,23]. However, the reality of the boost that the last-mile deliveries associated with a rise in e-trade shops provides in the ecosystem is indisputable. Egypt's Prime Minister said that Egypt must utilise environmental sustainability standards in nationwide investments and developments. The Prime Minister remarked upon the novel environment agenda, "Towards 2030: Agenda for a Greener Med", which aims to combine energies to fight climate transformation, hasten the green and digital revolution, prevent compulsory displacement and illegal immigration, as well as encourage peace in the Mediterranean region. In this regard, he evaluated several of the significant projects initiated by Egypt and tested through different studies [24,25] and the attempts that have been made to deal with the complicated disputes of climate shift, involving the issuing of green bonds for the original plan in the Middle East and North Africa. Egypt has additionally succeeded in preparing the first public climate change tactic in a meeting with the involved and civilian organisations.

Technological developments are essential in numerous techniques, with academics in multiple disciplines pursuing the development and implementation of new sharing

economy technologies that are more efficient, effective and that deliver more optimum added value [26,27]. A superior quality example of the sharing economy is the improvements in last-mile platforms technology that have occurred in recent years. This sharing economy machinery aims to support people in transporting goods and individuals across different locations to increase users' satisfaction in reaching their preferred and tailored destination [28]. Regarding the practicality of the sharing economy, specifically in last-mile application technology, the implementation rate is relatively slow and may be deterred by unknown factors that impede adoption decisions [29]. As proposed in epidemic theory, establishing a sharing economy in any given society can be convoluted and prolonged [30]. As an application of the sharing economy, the last-mile logistics sector deals with the social and environmental barriers to starting operations while conforming with environmental and social specifications [31]. Some users may want to take the first move adopt early, while others may prefer to play it safe and not rely on new technologies [32]. Others may be contemplating their decisions because they have controlled resources or because the assumed advantages are not adequately compelling [33]. This advances our research questions: (I) What are the fundamental relationships between environmental and societal factors and societies' resistance to last-mile adoption? (II) Which of the determining factors inside the model framework are highly significant and what is the nature of the relationships between these factors?

The motivation for writing this paper is the increase in the pollution caused by transportation and the high cost of transportation. The fact that Egypt is one of the developing countries, and the huge issue of the currency pricing problem, lead to a huge fluctuation in all services. The researchers were motivated to pursue this research in order to reduce transportation costs and apply the sharing economy concept through this application. In addition, another motive is that Alexandria is a major metropolitan North African city that can be adopted as any other African major city, as this will lead to a huge reduction in costs and pollution and increase the efficiency of services.

The knowledge gap remains broad despite the numerous observations and studies related to last-mile delivery in the literature. The current findings highlight the enhancements this technology can bring to societies and individual parcel delivery performance. Nevertheless, to our understanding, few studies have tried to distinguish its barriers to adoption. Thus, this research aims to meet this small but essential gap, bringing the last-mile one step closer to being put into practice. The researchers tried to recognise the factors driving the adoption of last-mile delivery applications and emphasise the main barriers stopping societies from adopting last-mile delivery applications. The researchers also present provisions and modifications for these interests based on the findings. The findings may help to strengthen the assertions of upcoming theories. We believe many aspects in our investigation and assessment procedure determine their pertinence. It is recognised that, although there are review papers that have studied the barriers, we believe that, from the point of view of society's acceptance and stakeholders' integration, there is still demand for more concentrated consideration. The researchers outline the obstacles in the model framework and understand those obstructions as unique to the area. The study also offers provisions and improvements for these concerns based on the findings.

The rest of this paper is constructed as follows. In the literature review, we recapitulate the significant current works. Then, we identify some essential phrases and our theoretical framework, based on which we describe our hypotheses in the subsequent section. The research method is also explained, alongside its validity. Following this, we describe and discuss our results. The last section completes the paper, including a discussion of guidelines for potential research.

## 2. The MAY-D System

Many studies tackle the sharing economy concept and address many questions related to applying a platform [34–36]. In addition, realising the constraints and barriers to using such an application in developing countries is the main objective of this research [37–39].

This platform is mainly for exchanging passenger or package transportation services between different areas. Each user is set up with an e-Wallet. The offer owner accepts the selected demand (max 3). The demand owner takes or rejects the offer. If taken and an agreement is drawn up, each offer can be given a maximum of 3 arrangements. Any group user can post either a demand or a request for transportation. When the demanding owner enters the car, he confirms the start of his agreement, and the driver starts once he can see the confirmation. Once the driver arrives, he confirms the stop, which is also given to the demanding owner. Simultaneously, a ticket is transferred from his e-Wallet. The demand owner can file an appeal in the case of abuse.

Many initiatives have been carried out for the logistics sector through the sharing economy, especially those related to applications which create value for all stakeholders, such as resource coordination [6], better logistics service for customers [40,41], and the ability to optimise costs [42,43] and reduce environmental pollution [44,45]. Moreover, the study seeks to answer the call from Valente, Patrus [38] and Tu, Aljumah [46] to study the social, environmental, and technological barriers that could impact the adoption of last-mile logistics applications in developing countries.

## 3. Technology Acceptance Model

Numerous theoretical models have been used to analyse the user acceptance and usage behaviour of developing technologies [47]. While many models feature perceived ease of use as an element of approval, the Technology Acceptance Model (TAM) is the exceptionally commonly applied user acceptance and usage model. TAM was adapted from the Theory of Reasoned Action (TRA) [48,49].

TAM proposes that two specific beliefs—perceived ease of use and perceived usefulness—control one's behavioural intention to use a technology, which have been associated with subsequent behaviour. Boldness in regard to using technology was mislaid because of the incomplete mediation of the influence of beliefs on the intention by attitude, a weak direct link between perceived usefulness and attitude, and a strong direct link between perceived usefulness and purpose. This was described as deriving from people aiming to use a technology because it was beneficial even though it did not have a progressive impact regarding its use. The absence of attitude eases the enhanced understanding of the effect of perceived ease of use and perceived usefulness on the key dependent variable of interest.

Furthermore, TAM suggests that perceived usefulness will be affected by perceived ease of use because other things being equal, the simpler a technology is to use, the more beneficial it can be. TAM proposes that the impact of external variables on intent is resolved by fundamental beliefs (i.e., perceived ease of use and usefulness). TAM has received massive empirical support through validations, applications, and replications [11,50,51].

Researchers and practitioners indicate that TAM is robust across time, settings, populations, and technologies. Perceived ease of use is the extent to which a person considers that using technology will liberate them from needing to make more effort. Perceived ease of use is a hypothesis connected to an individual's evaluation of the effort embraced in using the system. It is worth noting that other theoretical perspectives studying user acceptance have also used similar constructs; Thompson et al. [52] used a construct called "complexity" and Moore and Benbasat [53] employ a construct called "ease of use".

## 4. Barriers' Impact on Last-Mile Application

Many substantial frameworks determine the progression and rapidity of a society's technology adoption [54]. Firstly, the societal perspective describes the conventional and confidential arrangements of the developers, the importance of innovation, the capacity and volume of the society, insufficient assets, and communication development [55]. Secondly, the environmental perspective is the blend of additional factors that impact the application from the exterior perspective. This comprises a market structure, competition, external pressure/support accessible for implementing innovative technologies, and government policies. These predecessors act together to impact technology adoption



decisions [55]. These factors comprise all barriers from formerly cited assumptions. This assumption shows that the previously mentioned barriers retain effective interactions with technological endurance and momentum in implementation from both the users and developers' perspectives.

### 4.1. Environmental Barriers

The last barrier to be considered is anxiety about external individuals who do not participate enthusiastically in the chain but who affect supply chain events, such as authorities, organisations, and businesses [15]. The scarcity of administrative and business procedures diminishes the adoption momentum and hinders stakeholders from participating in last-mile applications [16]. Firms of all proportions still examine its applications, repeatedly implementing a pause-and-find methodology instead of standing as the initial transporters. To increase the speed of the participation of peripheral stakeholders, facilitators, such as governmental trade and industry assistance and aid, are mandatory [18]. Concentrating on firms with last-mile projects and offering legal support, financial assistance, seminars, training curricula, and similar should decrease businesses' resistance to last-mile implementation. Finally, a superior quality infrastructure is required and should be the ultimate significant factor in modern high-tech conversions. The existing technology foundation is not sustainable yet effective. Here, continuous, high-paced internet and electrical energy are essential elements that encourage usability [56].

### 4.2. Societal Barriers

One study demonstrates that developers' assistance and societal enthusiasm are essential to implementing last-mile applications. Furthermore, the extent of the application by developers, users' technology perception, and the absence of information modify not only the rate of implementation but also the enthusiasm for initiating last-mile adoption [57]. Some users face commitment limitations, while long-term commitment is fundamental to the efficacious adoption of technology [58]. The numerous activities of the stakeholders that influence the conclusions of technology implementation involve their assistance for monetary and technical support, eliminating obstacles, solving complications, persuading all stakeholders to yield a portion in the development, and allotting their revelation [59]. Last-mile endurance may also be associated with incompetence regarding similar applications inside society [60]. The current expansion of technology has broadened the disparity between the needs and resources on both the developer and user sides. To comprehend the full capability of this technology, society must be experienced in both IT and regular practices [61]. Thus, training and encouraging the community to use last-mile applications is difficult. Without enough qualified users and developers, businesses may not harvest the full advantages presented by last-mile and, therefore, may not want to implement it [62].

The challenges associated with collaboration between stakeholders can be an excruciating process, prominent in the endurance of some of society as not all participants view the worth of change in the same traditional way [63]. Additionally, implementing modern technology may alter the prevailing societal culture and, consequently, necessitate the establishment of new roles, responsibilities, proficiencies or capacities in order to organise and support numerous characteristics [64]. Monopolistic supremacy may, correspondingly, be a barrier for novel adopters. Last-mile developers hold significant power when implementing the application. Monopolies transpire when one business positions other firms in a differentiating position by manipulating the majority of the supply for a product or service in the arena [65].

In supply chain grids, the distinct businesses' entrances for technology implementation should be considered with other firms' decisions within a network, together with their societal qualities. Due to this assortment through firms, such as diverse network sizes, previous beliefs, and the quantity of data studied, each firm reaches a conclusion for implementation at different periods [66]. Consequently, they may not wish to share information and may, as a substitute, enforce excessive fortification [67]. Internal information is highly vulnerable

and hidden from outsiders. In order to break down this resistance, companies must seize enough necessary information [68]. In addition, each society possesses a unique culture, and cultural variations may be points of disagreement in supply chain relationships [69].

### 4.3. Technological Barriers

Security and vulnerability originated as barriers to implementation from the technological perspective, as trade-offs must be made between security and performance [70]. Complicity is tranquil and conceivable over compromise among participants. It is certain that privacy anxieties are of major significance [71]. Furthermore, the confrontation could arise from an absence of tuning of authentic practices. The absence of tuning standards by developers is a barrier to implementation intentions. Additionally, the nonexistence of a uniform computer language impedes application developers when they determine that platforms cannot link without support [72]. Other factors to consider are scalability and rapidity, as well as the application's capability in accomplishing an operation and achieving its intentions in a good timeframe [73].

Strong backing and understanding must be provided by associating stakeholders (users and developers) to put this technology into practice. Consequently, the implementation structure of the technology complements an additional layer of complications for application adaptation [74]. Last-mile technology has only been freshly uncovered, and its innovation remains an unsettled subject that grounds complications. The number of fruitful implementations is inadequate for the goals of adoption. Therefore, it might not be the case that the technology will mysteriously elevate adaptation. Adopters are still uncertain, inferior performers are accompanying investigations and progress regarding this technology, and limitations forced by stakeholders delay the evolution stages for the adoption of last-mile [75].

Another feature that should be measured when applying innovative knowledge is its compatibility. Stakeholders must furthermore procure and develop last-mile applications that are well-suited to their daily lifestyle arrangements or transmute their contemporary understandings to be fit-harmonised with last-mile application technology [76]. Financial factors stand out as one of the core barriers to adoption. The cost of implementing last-mile is not convincing, which may hamper the backing and commitment of the stakeholders [77]. Implementation expenses may diverge because of numerous profound influences, including hardware, software systems, enrollment, and in-community preparation. Last-mile is rumoured to be a technology with low simple asset costs, and it brings compensation in terms of user cost reduction [59].

From the factors mentioned in the above literature, some of the factors were identified for assessment. These factors are the most cited features in the previous literature related to the barriers related to implementation issues.

## 5. Hypotheses Development

### 5.1. Environmental Hypothesis

Backing from the government is significant in encouraging the implementation of innovative technologies. The discernment of the nonexistence of governmental support, in the arrangement of funding or compassionate policies, prevents societies from considering implementing the technology. Perceptions and procedures such as cryptanalytic signs and intelligent agreements have been obtainable, despite the absence of regulations. Users and developers are ambiguous about the law of the last-mile; for example, it is still undecided who will function as an authority in opposition circumstances. Resourceful technological infrastructure is essential for societies to be proficient enough to understand such technology's advantages. For example, continuous and elevated-velocity internet and electrical energy are significant factors. Therefore, we hypothesise the following:

**Hypothesis 1 (H1).** *A greater restriction on government support boosts societies' resistance to last-mile.*

**Hypothesis 2 (H2).** *A better understood restriction on current regulations and legitimate contexts increases societies' opposition to last-mile.*

**Hypothesis 3 (H3).** *A greater supposed restriction of an effective technological foundation increases societies' opposition to last-mile.*

*5.2. Societal Hypothesis*

When a novel form of technology is announced into society, developers and application users are weighty decision makers with value in implementation. However, this level of acquaintance with the technology is associated with their return; decision makers' minds turn to performance concerns when combatting uncertainty. The last-mile system is novel and acknowledged as a superior technology. Since its commencement, few stakeholders have influenced satisfactory proficiency or professional knowledge to practice the technology. One essential is having specialised data on technological knowledge in order to comprehend the possible expenses and gains of this innovative technology. Last-mile application developers gain much influence when designing systems, which positions users in an unfair situation. In this framework, last-mile platform developers may try to lock in their customers. For example, implementation requires a substantial infrastructure investment, making it problematic to switch to another platform in the future. An active contribution is compulsory from all associated parties. Although communication is significant, it is also stimulating because users must be thoughtful about sharing personal information. With this logic, each party attempts to ensure that admittance is offered only to data or information pertinent to the application while preserving honest relationships. We recommend the following hypotheses:

**Hypothesis 4 (H4).** *A lower collaboration effort required for collaboration between stakeholders increases societies' resistance to last-mile.*

**Hypothesis 5 (H5).** *Lower expertise and technical knowledge increases societies' resistance to last-mile.*

**Hypothesis 6 (H6).** *Lower technological knowledge and awareness increases societies' resistance to last-mile.*

*5.3. Technological Hypothesis*

This paper describes complexity as the quantity of struggle essential in recognising a technology from an industry point of view. The more complicated a technology is, the less possible it is for it to be rapidly executed. When technology is problematic, its implementation is often unrestrained or postponed. Therefore, we recommend that this last-mile application complication has a definite impact on users' resistance to using it. Technological development is the extent to which last-mile technology has been enjoyed since its initial presence. It is naturally simpler for a society to adopt a technology if the conclusion has reached maturation and is employed extensively in different communities, outstanding for its wealth and deep understanding. In other words, naivety exhausts the appraisal mechanism.

From a modernisation perspective, we describe compatibility as the level to which some knowledge resembles a society's legacy classification, procedures, data technology infrastructure, and additional associations with which it is predicted to thrive. It is simpler for a corporation to operate a technology if it has a superior compatibility degree. Last-mile application is exceptionally associated with different technologies. The velocity of the operations compensates for the protection aspects. At this point, scalability refers to the acceleration and size of processes. The technology has been complained about, since it was initially presented, for its scalability problems, with numerous academics remarking that, were it not for these restrictions, last-mile applications could now have a diverse position. Therefore, we recommend the following hypotheses for the technology framework:

**Hypothesis 7 (H7).** *Elevated technological complication boosts societies' resistance to last-mile.*

**Hypothesis 8 (H8).** *Reduced technological scalability boosts societies' resistance to last-mile.*

**Hypothesis 9 (H9).** *Elevated technological protection and confidentiality concerns boost societies' resistance to last-mile.*

**Hypothesis 10 (H10).** *Elevated execution expenses boost societies' resistance to last-mile.*

Based on the above discussion, the research model has been developed and presented in Figure 1.

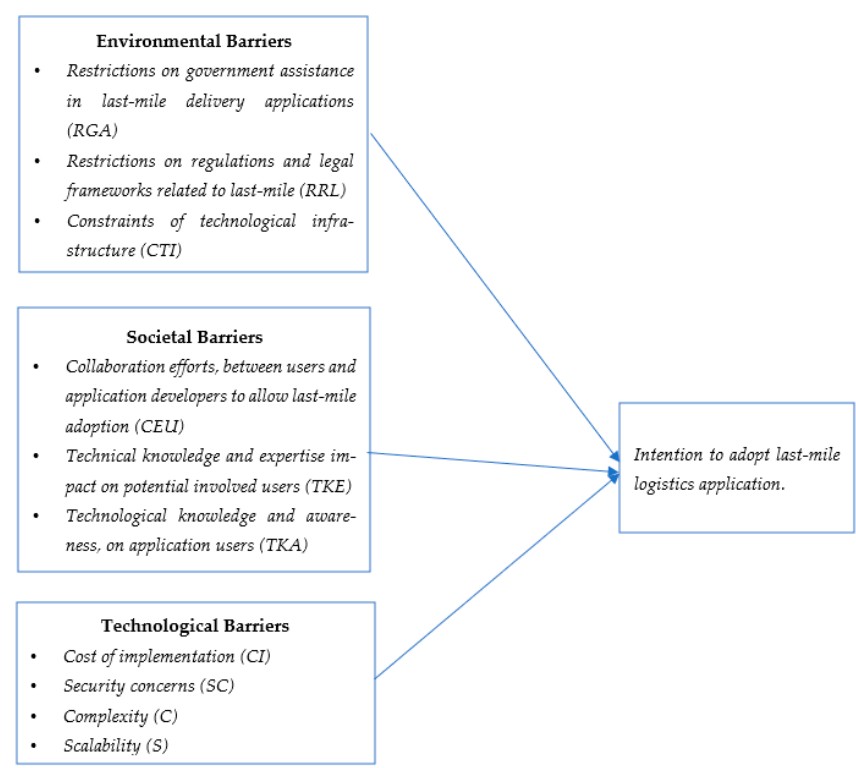

**Figure 1.** The research model. Source: this research.

## 6. Research Methodology

### 6.1. Study Context

Data were collected from users in the Egyptian market, more specifically Alexandria as a major metropolitan city in the Egyptian region and the North African region, as well as being populated by more than 5.5 million citizens in 2023, with a growth rate of almost 2% per year [78]. The focus on a developing country (Egypt) can be broadened, as it is one of the most important economic zones in the southern Mediterranean Sea, and the increased number of inhabitants will be a much stronger incentive for the concept of the sharing economy [79]. In addition, this study builds upon previous research conducted in Egypt, specifically focusing on data collection from three major cities: Cairo, Alexandria, and Giza. As previously stated, the study aims to implement a new application. Consequently, it is advisable to initially implement it on a small scale before considering its applicability on a larger scale. Therefore, the study was conducted in Alexandria. Egypt has formulated a set of eight sustainable goals in alignment with the United Nations Sustainable Development Goals (SDGs). These goals encompass various aspects, one of which is to reduce the overall emissions at all levels in different sectors [80]. Moreover, the implementation of sustainability objectives by the government has prompted organisations across various sectors to adopt diverse practices aimed at identifying optimal solutions for reducing overall

emissions, achieving optimal performance, and to increase the individual's knowledge and awareness. Therefore, developing countries' governments have urged looking into new emerging ideas, like the sharing economy and last-mile delivery, in order to reduce the number of traveling vehicles with small parcels to the final destinations to reduce the overall emissions [81]. Furthermore, failure to adopt these new concepts due to the lack of societal and environmental awareness is a barrier to implementing such ideas in developing environments (Egyptian environment) [17]. Because of the lack of societal and ecological awareness regarding this adoption, this study opted to pre-test and pilot the data collection instrument in Alexandria, Egypt, before the main study.

### 6.2. Sample and Procedures

Non-probability sampling techniques (self-selecting sampling and snowball sampling) were implemented in this research for both the pilot and main study [82]. Although these techniques were condemned as biased and subjective, the probability of selection is not uniformly distributed among all members of a group, and cooperation may be lacking [83]. They were applied in this research as there was no sampling frame available (there are no available application adoption databases in Egypt) [84]. Data were collected from resident users in Alexandria, Egypt; participants were individuals with different backgrounds and genders interested in adopting new technologies [85]. The selection of these residents is based on the knowledge that they are a part of the tested society where the adoption will take place, in addition to their ability to provide reliable information [86]. Users of this proposed application are responsible for achieving a massive portion of the concept's adoption and ensuring the efficient use of it [87]. In addition, they are responsible for evaluating whether it will be quickly adopted or whether there are other points to be considered while adopting this concept and its application [28].

The sample size is proposed to be 5 or 10 observations per measurement [88]. Nevertheless, ten observations per measurement increase the accuracy of the results [89]. Applying the rule of ten, the sample size targeted for the pilot and the main study in this research will exceed 270. To accomplish this target, the self-administered questionnaire was distributed online through emails to fifty participants for the pilot study. The items of the questionnaire have been adopted from previous studies, as indicated in Table 1. A total of 1060 separate questionnaires were distributed for the main study, with a total of 50 and 1000 gathered valid questionnaires for the pilot and main study, respectively. This means the response rate was 100% for the pilot and 100% for the main study.

### 6.3. Research Instruments

This study employs scales derived from prior research studies in order to enhance the validity and reliability of the findings. The questionnaire items were not directly related to the adoption of last-mile logistics; however, they were testing the adaptability of applications. Consequently, the researchers conducted a pre-test and pilot study to assess the appropriateness of these items for measuring last-mile logistics. The items within the questionnaires were assessed using a seven-point Likert scale, as recommended by Hair [90], for the purpose of enhancing accuracy and precision. Additionally, this approach was employed to improve the reliability of the measurements, as suggested by Dawes [91]. The questionnaire is structured into five primary sections. The first one is related to gathering demographic information, including Gender, Age, Geographical Location, Period Staying in the Area, and Educational Level. The remaining sections include the research variables Environmental Barriers, Societal Barriers, Technological Barriers, and Last-mile Logistics Adoption. Environmental Barriers will be tested through three sub-variables (11 items), while Societal Barriers will be measured through three sub-variables (12 items), whereas Technological Barriers will be tested through four sub-variables (14 items), and last-mile logistics adoption will be measured through four items. The research items are indicated in Table 1.

**Table 1.** The main variable, sub variables, and measuring questions.

| Main Variables | Sub-Variables | Questions Assessing the Variables |
|---|---|---|
| Environmental Barriers | Restrictions to government assistance in last-mile delivery applications (RGA) [92] | The government has not supplied incentives to encourage the implementation of last-mile application in Egypt. |
| | | The Egyptian government does not enthusiastically encourage last-mile application implementation. |
| | | The Egyptian government has not launched significant legislations to boost last-mile application implementation. |
| | | There is no backing offered by the Egyptian government regarding last-mile implementation. |
| | Restrictions on regulations and legal frameworks related to last-mile (RRL) [70,93,94] | The governing body (Egyptian cabinet) is not so far well determined to deal with last-mile issues. |
| | | There is no plan for adjustments in policies that would interfere with our practices of last-mile application in the future. |
| | | There is no authority to solve disputes between users and developers. |
| | | Legal structures do not satisfactorily safeguard users from problems on last-mile platforms. |
| | Constraints of technological infrastructure (CTI) [95–97] | The contemporary technological structure is not adequate for last-mile implementation. |
| | | The existing internet service is not efficient enough for last-mile implementation. |
| | | There is not satisfactory access to last-mile technology. |
| Societal Barriers | Collaboration efforts, between users and application developers, to allow last-mile adoption (CEU) [98,99] | Is not easy. |
| | | Is challenging. |
| | | Demands a lot of intellectual effort. |
| | Technical knowledge and expertise impact on potential involved users (TKE) [100,101] | Has the relevant technical knowledge about last-mile technology. |
| | | Users are qualified to use last-mile applications. |
| | | Has interest in projects related to last-mile technology. |
| | | Is familiar with this type of technology and its applications. |
| | Technological knowledge and awareness on the part of application users (TKA) [102] | Does not recognise last-mile as a competitive weapon. |
| | | Does not recognise last-mile as an instrument to improve income and lifestyle. |
| | | Does not recognise last-mile as a tool to increase the usage of unused space in vehicles. |
| | | Does not recognise the potentials involved in participating in last-mile application. |
| | | Does not believe last-mile contributes significantly to their life or financial welfare. |

**Table 1.** *Cont.*

| Main Variables | Sub-Variables | Questions Assessing the Variables |
|---|---|---|
| Technological Barriers | Cost of implementation (CI) [103] | Increase the user's income generated from their assets. |
| | | Are expensive due to trial-and-error. |
| | | Require high up-front investment costs. |
| | Security concerns (SC) [104] | Do not feel secure in providing sensitive information when working with last-mile applications. |
| | | Do not feel secure sending and/or uploading sensitive information to the platform. |
| | | Do not feel safe sending precious parcels through last-mile applications. |
| | | Do not feel that last-mile is a safe platform for operating businesses with sensitive information and cargo overall. |
| | Complexity (C) [105] | Last-mile is conceptually difficult to understand from a business perspective. |
| | | Last-mile is conceptually difficult to understand from a technical perspective. |
| | | When using last-mile technology, it is difficult to resolve transactional errors. |
| | | Using last-mile technology is difficult. |
| | Scalability (S) [95,106] | The speed of operation of last-mile will never be enormous. |
| | | Last-mile operation size will never be huge as users will be repelled from implementing this concept. |
| | | Overall expected operation size and speed are to be problematic aspects. |
| The Intention to Adopt the Sharing Economy Concept (Last-Mile) in Alexandria, Egypt | Regarding our stance on last-mile application technology [107,108] | Will NOT adopt last-mile application unless it proves beneficial for us. |
| | | Needs to clarify some queries regarding last-mile and justify adopting last-mile application. |
| | | Is unlikely to use last-mile application soon. |
| | | Believes that last-mile application is not for Alexandria city. |

### 6.4. Descriptive Analysis

Descriptive statistics is a methodological tool employed to elucidate and provide a comprehensive understanding of the characteristics inherent in a specific dataset. It accomplishes this by offering concise summaries of the participants and the manner in which the process of diversification was implemented to ensure the selection of a representative sample from the population being investigated [109]. The data presented in this study are depicted through frequency tables, which display both the count and percentage of participants who responded to the questionnaire within each category. Table 1 shows the demographic characteristics; it is important to mention that the younger in this table refers to millennials and generation Z while the elder refers to the silent generations of baby boomers and generation X. Also, we have divided Alexandria into two main areas: Agamy surrounding Amerya, Borg Alarab, and the west Alexandria area, and Montaza surrounding the Gomrok, middle, and Montaza areas. Finally, regarding education level, postgrad refers to master's and doctorate degree holders, and undergrad refers to high schoolers and bachelor's degree holders. Table 2 shows the sample characteristics, and Table 3 shows the Mean, Median, Mode, and Std. Deviation for the research items.

**Table 2.** Sample characteristics.

|  |  | Frequency | Percent |
|---|---|---|---|
| Gender | Male | 623 | 58.8 |
|  | Female | 437 | 41.2 |
| Age | Younger | 218 | 20.6 |
|  | Elder | 842 | 79.4 |
| Geographical Location | Agamy_surround | 778 | 73.4 |
|  | Montaza_suround | 282 | 26.6 |
| Staying at Area | less than 1 year | 40 | 3.8 |
|  | 1–3 years | 98 | 9.2 |
|  | 6–8 years | 315 | 29.7 |
|  | more than 8 years | 607 | 57.3 |
| Educational Level | Postgrad | 464 | 43.8 |
|  | Undergrad | 596 | 56.2 |

Source: this research.

**Table 3.** Descriptive statistics.

|  | Mean | Median | Mode | Std. Deviation |
|---|---|---|---|---|
| RGA | 2.267 | 2.0000 | 1.25 | 1.24300 |
| RRL | 2.7493 | 2.2500 | 1.25 | 1.65130 |
| CTI | 5.5349 | 6.0000 | 6.33 | 1.42719 |
| CEU | 2.8733 | 2.2500 | 1.00 | 1.63644 |
| TKE | 5.663 | 6.0000 | 6.50 | 1.31797 |
| TKA | 2.6174 | 2.0000 | 1.20 | 1.60580 |
| CI | 2.4711 | 2.0000 | 1.33 | 1.54011 |
| SC | 5.1061 | 6.2500 | 6.25 | 1.93885 |
| C | 2.5012 | 1.7500 | 1.50 | 1.68513 |
| S | 2.8352 | 1.6667 | 1.33 | 1.91288 |
| LM | 6.041 | 6.5000 | 6.50 | 1.28222 |

Source: this research.

## 7. The Research Findings

### 7.1. Pre-Test

Face validity will be used through expert review (five filed from academics and four from practitioners) as research variables extracted from the literature and to guarantee content validity [110,111]. To ensure that the context of the questionnaire was measuring what it was intended to measure, the experts were requested to evaluate and assess the questionnaire and ensure that it was understandable, readable, answerable, and not complicated. The selection of experts was based on their professional positions and years of experience. The researchers focused on selecting experts whose expertise would enhance the content of the questionnaire and to add valuable advice. Moreover, due to the application of the study in Egypt, the targeted sample received the questionnaires in Arabic to be appropriate to Egyptian culture and the nature of the research. So, to test the translation accuracy, the back-translation method was used in this study [112]. The researchers expected to have expert feedback and comments, and the changes were made based on their opinions.

### 7.2. Pilot Study

The primary goals of this pilot study were to verify the validity of the questionnaires, determine whether they were correctly worded and easily understood by respondents, and assess their reliability. Two hundred and thirteen participants from the target sample are used in this study, which measures the reliability and validity of the questionnaire through SPSS. Factor loadings, composite reliability and average variance extracted were obtained to test the reliability and validity, and to modify or delete elements or statements. The results of the pilot study are indicated in Table 4, which illustrates the validity and reliability test and presents factor loading, composite reliability, and AVE with the threshold values of 0.5, 0.7, and 0.5, [113] respectively.

**Table 4.** Pilot study reliability and validity.

| Latent Variable/Construct | | Items | Factor Loading | Composite Reliability | Composite Reliability after Items Were Deleted | AVE | AVE after Items Were Deleted |
|---|---|---|---|---|---|---|---|
| Environmental Barriers | Restrictions on government assistance in last-mile delivery applications | RGA1 | 0.893 | 0.934 | - | 0.83 | - |
| | | RGA2 | 0.929 | | | | |
| | | RGA3 | 0.948 | | | | |
| | | RGA4 | 0.889 | | | | |
| | Restrictions on regulations and legal frameworks related to last-mile | RRL1 | 0.931 | 0.923 | - | 0.81 | - |
| | | RRL2 | 0.944 | | | | |
| | | RRL3 | 0.943 | | | | |
| | | RRL4 | 0.780 | | | | |
| | Constraints of technological infrastructure | CTI1 | 0.924 | 0.872 | - | 0.79 | - |
| | | CTI2 | 0.834 | | | | |
| | | CTI3 | 0.918 | | | | |
| Societal Barriers | Collaboration efforts between users and application developers to allow last-mile adoption | CEU1 | 0.844 | 0.76 | 0.86 | 0.61 | 0.77 |
| | | CEU2 | 0.903 | | | | |
| | | CEU3 | 0.876 | | | | |
| | | CEU4 | 0.404 (Deleted) | | | | |
| | Technical knowledge and expertise impact on potentially involved users | TKE1 | 0.854 | 0.88 | - | 0.74 | - |
| | | TKE2 | 0.923 | | | | |
| | | TKE3 | 0.893 | | | | |
| | | TKE4 | 0.783 | | | | |
| | Technological knowledge and awareness in application users | TKA1 | 0.868 | 0.90 | - | 0.73 | - |
| | | TKA2 | 0.872 | | | | |
| | | TKA3 | 0.859 | | | | |
| | | TKA4 | 0.881 | | | | |
| | | TKA5 | 0.796 | | | | |
| Technological Barriers | Cost of implementation | CI1 | 0.844 | 0.80 | - | 0.71 | - |
| | | CI2 | 0.834 | | | | |
| | | CI3 | 0.861 | | | | |
| | Security concerns | SC1 | 0.862 | 0.88 | - | 0.74 | - |
| | | SC2 | 0.874 | | | | |
| | | SC3 | 0.928 | | | | |
| | | SC4 | 0.785 | | | | |

**Table 4.** *Cont.*

| Latent Variable/Construct | | Items | Factor Loading | Composite Reliability | Composite Reliability after Items Were Deleted | AVE | AVE after Items Were Deleted |
|---|---|---|---|---|---|---|---|
| | Complexity | C1 | 0.918 | 0.91 | - | 0.80 | - |
| | | C2 | 0.898 | | | | |
| | | C3 | 0.833 | | | | |
| | | C4 | 0.940 | | | | |
| | Scalability | S1 | 0.895 | 0.88 | - | 0.81 | - |
| | | S2 | 0.943 | | | | |
| | | S3 | 0.865 | | | | |
| Last-Mile | | LM1 | 0.773 | 0.80 | 0.85 | 0.57 | 0.69 |
| | | LM2 | 0.816 | | | | |
| | | LM3 | 0.341 (Deleted) | | | | |
| | | LM4 | 0.855 | | | | |
| | | LM5 | 0.865 | | | | |

Source: this research.

### 7.3. Non-Response Bias and Common Method Bias

To ensure that no single factor can account for 50% or more of the total variance, the common method bias technique was calculated using SPSS [114]. The results of the common method bias test, Harmon's one factor test, presented the fact that no single item explained more than 50% of the total variance [115].

For non-response bias, the main objective is to prevent a significant difference between early and late responses [116,117]. Levene's test results showed a non-significant *p*-value, which indicates that there is no difference between the early and late responses, as indicated in Table 5.

**Table 5.** Main study reliability and validity.

| Latent Variable/Construct | | Items | Factor Loading | Composite Reliability | AVE |
|---|---|---|---|---|---|
| Environmental Barriers | Restrictions on government assistance in last-mile delivery applications | RGA1 | 0.944 | 0.93 | 0.845 |
| | | RGA2 | 0.907 | | |
| | | RGA3 | 0.966 | | |
| | | RGA4 | 0.855 | | |
| | Restrictions on regulations and legal frameworks related to last-mile | RRL1 | 0.939 | 0.94 | 0.849 |
| | | RRL2 | 0.923 | | |
| | | RRL3 | 0.941 | | |
| | | RRL4 | 0.882 | | |
| | Constraints of technological infrastructure | CTI1 | 0.953 | 0.91 | 0.853 |
| | | CTI2 | 0.886 | | |
| | | CTI3 | 0.931 | | |
| Societal Barriers | Collaboration efforts between users and application developers to allow last-mile adoption | CEU1 | 0.943 | 0.94 | 0.887 |
| | | CEU2 | 0.935 | | |
| | | CEU3 | 0.948 | | |

**Table 5.** *Cont.*

| Latent Variable/Construct | | Items | Factor Loading | Composite Reliability | AVE |
|---|---|---|---|---|---|
| | Technical knowledge and expertise impact on potentially involved users | TKE1 | 0.908 | 0.90 | 0.767 |
| | | TKE2 | 0.906 | | |
| | | TKE3 | 0.913 | | |
| | | TKE4 | 0.768 | | |
| | Technological knowledge and awareness in application users | TKA1 | 0.947 | 0.95 | 0.836 |
| | | TKA2 | 0.942 | | |
| | | TKA3 | 0.908 | | |
| | | TKA4 | 0.938 | | |
| | | TKA5 | 0.833 | | |
| Technological Barriers | Cost of implementation | CI1 | 0.943 | 0.87 | 0.799 |
| | | CI2 | 0.942 | | |
| | | CI3 | 0.787 | | |
| | Security concerns | SC1 | 0.953 | 0.93 | 0.841 |
| | | SC2 | 0.916 | | |
| | | SC3 | 0.944 | | |
| | | SC4 | 0.854 | | |
| | Complexity | C1 | 0.967 | 0.92 | 0.828 |
| | | C2 | 0.965 | | |
| | | C3 | 0.954 | | |
| | | C4 | 0.731 | | |
| | Scalability | S1 | 0.871 | 0.89 | 0.786 |
| | | S2 | 0.866 | | |
| | | S3 | 0.923 | | |
| Last-Mile | | LM1 | 0.930 | 0.93 | 0.827 |
| | | LM2 | 0.942 | | |
| | | LM3 | 0.955 | | |
| | | LM4 | 0.804 | | |

Source: this research.

*7.4. Main Study*

The research model has been tested using CB-SEM. The use of this technique to analyse these data was due to many reasons. Firstly, the results are widely comprehensive and include all the values needed. Secondly, it makes simultaneous testing for all the research variables possible. Thirdly, its ability to handle complex models and data that do not follow a normal distribution [118]. Hair Jr, Howard [119] illustrated that factor loadings, composite reliability, and AVE are needed to evaluate the model reliability and validity before conducting the hypotheses testing. Table 6 indicates that all items meet the threshold values of 0.5, 0.7, and 0.5, respectively. Moreover, in Table 7 the discriminant validity is extracted through calculating the correlation matrix and contrasting the square root of AVE of each construct when the square root of AVE is higher than the correlation value between the variables as indicated in Table 8. The research results have been extracted using SMART PLS, as the study has more than six variables [120].

**Table 6.** Independent *t*-test.

| | | t-Value | Df | Mean Difference | Standard Error | 95% Confidence Interval | |
|---|---|---|---|---|---|---|---|
| | | | | | | Lower Limit | Upper Limit |
| Environmental Barriers | RGA | −0.870 | 1058 | −0.06643 | 0.07637 | −0.21629 | 0.08342 |
| | RRL | −0.358 | 1058 | −0.03631 | 0.10149 | −0.23545 | 0.16283 |
| | CTI | −0.045 | 1058 | −0.00394 | 0.08772 | −0.17606 | 0.16819 |
| Societal Barriers | CEU | −0.164 | 1058 | −0.01647 | 0.10058 | −0.21383 | 0.18088 |
| | TKE | 0.415 | 1058 | 0.03358 | 0.08100 | −0.12536 | 0.19252 |
| | TKA | 0.256 | 1058 | 0.02527 | 0.09869 | −0.16839 | 0.21893 |
| Technological Barriers | CI | −0.299 | 1058 | −0.02828 | 0.09466 | −0.21401 | 0.15745 |
| | SC | 0.510 | 1058 | 0.06081 | 0.11915 | −0.17299 | 0.29461 |
| | C | 0.205 | 1058 | 0.02120 | 0.10357 | −0.18202 | 0.22443 |
| | S | 0.235 | 1058 | 0.02763 | 0.11757 | −0.20306 | 0.25833 |
| Last-Mile | LM | 0.311 | 1058 | 0.02455 | 0.07880 | −0.13009 | 0.17918 |

Source: this research.

**Table 7.** Discriminant validity test.

| | C | CEU | CI | CTI | GL | HG | LM | RGA | RRL | S | SC | TKA |
|---|---|---|---|---|---|---|---|---|---|---|---|---|
| CEU | 0.125 | | | | | | | | | | | |
| CI | 0.369 | 0.152 | | | | | | | | | | |
| CTI | 0.285 | 0.614 | 0.355 | | | | | | | | | |
| GL | 0.110 | 0.058 | 0.068 | 0.132 | | | | | | | | |
| HG | 0.025 | 0.018 | 0.029 | 0.028 | 0.204 | | | | | | | |
| LM | 0.339 | 0.085 | 0.384 | 0.360 | 0.264 | 0.165 | | | | | | |
| RGA | 0.345 | 0.189 | 0.301 | 0.510 | 0.206 | 0.143 | 0.770 | | | | | |
| RRL | 0.189 | 0.325 | 0.205 | 0.784 | 0.172 | 0.076 | 0.579 | 0.710 | | | | |
| S | 0.787 | 0.090 | 0.268 | 0.191 | 0.074 | 0.026 | 0.418 | 0.326 | 0.170 | | | |
| SC | 0.775 | 0.057 | 0.673 | 0.187 | 0.080 | 0.019 | 0.254 | 0.235 | 0.069 | 0.591 | | |
| TKA | 0.329 | 0.316 | 0.782 | 0.298 | 0.032 | 0.022 | 0.307 | 0.260 | 0.105 | 0.195 | 0.493 | |
| TKE | 0.373 | 0.524 | 0.440 | 0.466 | 0.065 | 0.019 | 0.310 | 0.404 | 0.258 | 0.243 | 0.299 | 0.746 |

Source: This research.

**Table 8.** Hypotheses testing.

| IV | Dependent | B | *p*-Value | Decision |
|---|---|---|---|---|
| C | LM | −0.031 | 0.180 | Rejected |
| CEU | LM | −0.015 | 0.227 | Rejected |
| CI | LM | −0.173 | 0.000 | Supported |
| CTI | LM | −0.214 | 0.000 | Supported |
| RGA | LM | −0.475 | 0.000 | Supported |
| RRL | LM | −0.265 | 0.000 | Supported |
| S | LM | −0.282 | 0.000 | Supported |
| SC | LM | −0.153 | 0.000 | Supported |

**Table 8.** *Cont.*

| IV | Dependent | B | *p*-Value | Decision |
|---|---|---|---|---|
| TKA | LM | −0.113 | 0.002 | Supported |
| TKE | LM | −0.039 | 0.111 | Rejected |

Source: this research.

## 8. Results

The research model has been evaluated using CB-SEM; the results are illustrated in Table 7 and Figure 2. The results show that the first hypothesis, that a higher perceived constraint on government support increases societies' resistance to last-mile, is supported by the negative relationship between the two variables. As β = −0.475, *p*-value is 0.000. For the second hypothesis, a higher perceived constraint on existing regulations and legal frameworks increases societies' resistance to last-mile. The results show a negative relationship between the RRL and LM as β = −0.264, *p*-value is 0.000.

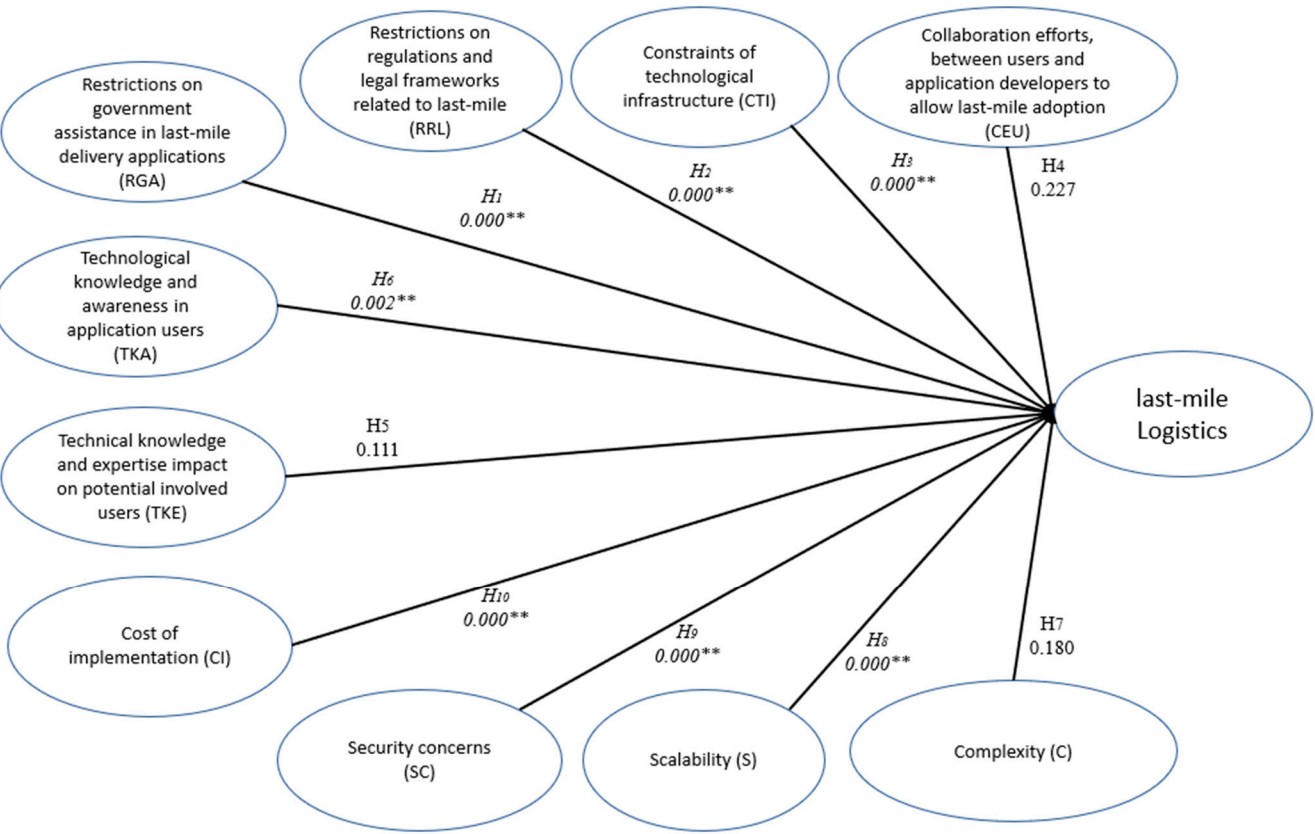

Note: **Means significance level at 1%.

**Figure 2.** SEM results. Source: this research.

Regarding the third hypothesis, a higher perceived constraint of an efficient technological infrastructure increases societies' resistance to last-mile. This hypothesis is supported by the fact that β = −0.213, *p*-value is 0.000. The first set of hypotheses related to Environmental Barriers is supported. For the Societal Barriers, the fourth hypothesis, that lower collaboration effort required for collaboration between stakeholders increases societies' resistance to last-mile, is not endorsed as β = −0.014, *p*-value is 0.226. In the same context, the fifth hypothesis is also unsupported; lower expertise and technical knowledge increase societies' resistance to last-mile (β = −0.038, *p*-value is 0.110). In contrast, the sixth hypothesis, that lower technological knowledge and awareness increase societies' resistance to last-mile, is supported by the fact that β = −0.112, *p*-value is 0.001. Regarding

the Technological Barriers, the seventh hypothesis, that higher technological complexity increases societies' resistance to last-mile, is unsupported ($\beta = -0.031$, *p*-value is 0.180). The eighth, ninth, and tenth hypotheses are all supported as $\beta = -0.282$, *p*-value is 0.000, $\beta = -0.153$, *p*-value is 0.000, and $\beta = -0.173$, *p*-value is 0.000, respectively. Figure 2 shows the overall model results.

## 9. Discussion

Environmental barriers include restrictions on government assistance in last-mile delivery applications, regulations and legal frameworks related to last-mile, and constraints on technological infrastructure. As the government supplying an incentive will encourage both users and developers to adopt last-mile technology, this will be carried out through adopting legislation and backing. This is achieved by improving both laws and policies to solve disputes. Also, the countries shall adopt new infrastructures as excellent and improved internet services allow easy access to last-mile technologies. Many last-mile delivery applications face these challenges when adapting to a new city and need to measure them in order to be capable of penetrating the new market [121]. This finding is in harmony with some research that supports the impact of RGA, RRL, and CTI in negative relation to societies adopting last-mile delivery, e.g., Boysen, Fedtke [84] examines both traditional and innovative last-mile strategies, with a particular focus on the decision-making challenges associated with implementing and managing each approach. Governmental policies should be designed to promote the adoption of this emerging technology as a means of addressing the associated challenges. El Moussaoui, Benbba [122] examined the primary obstacles encountered in last-mile logistics and explores potential strategies for enhancing its efficiency. The authors suggest that, in order to achieve an optimal last-mile model, it is imperative to focus on the development of technological infrastructure, supporting government and other related elements in the future.

Societal barriers also contain multiple restrictions in the adoption of last-mile delivery in major cities like Alexandria, Egypt. Firstly, collaboration efforts between users and application developers allow last-mile adoption. The collaboration effort between different users and between users and developers should be easy, not challenging, and should not need great intellectual effort. However, the ability of individuals to effectively engage with others may be hindered in the absence of a substantial degree of mutual trust. Certain individuals perceive collaboration as laborious, problematic, and burdensome. This results are consistent with Park, Park [123], who suggested that collaboration could be carried out under some conditions. Secondly, technical knowledge and expertise have an impact on the potentially involved users; thus, in order to be deemed eligible for the adoption and utilisation of last-mile application, users must possess a sufficient level of technical expertise. Furthermore, it is imperative that users exhibit a genuine interest in engaging with last-mile delivery applications. As Egypt is one of the developing countries, the utilisation of applications such as this may not rank among the primary concerns of the user base. Thirdly, in terms of the technological knowledge and awareness of application users, the provision of technical awareness is essential in order to cultivate an understanding of the strategic significance of last-mile as a competitive advantage for augmenting one's income and optimising the utilisation of underutilised space in vehicles, thereby enhancing the well-being of users. Third-world major cities have unique societies with different backgrounds and concepts from those of other European major cities, which may interfere with the adaptation of last-mile delivery applications [124]. This supports the hypothesis of the existence of negative relationships between CEU, TKE, and TKA, and the adaptation of last-mile delivery, which is supported by the findings of some researchers such as those of the authors of references [125–127].

The impact of technological barriers has been discussed in multiple research studies, such as in references [128,129]. The technological barriers' main players are cost of implementation, security concerns, complexity, and scalability. The first factor affecting this barrier is the cost of implementation, as the users will be aware that last-mile will improve

their asset value and will not be expensive but may require a high upfront investment. This result is consistent with Ranieri, Digiesi [130], who explored novel approaches in the realm of last-mile logistics, with a specific emphasis on the reduction in costs. Although the applicability cost could be higher and find a huge resistance, the knowledge of the application advantages would reduce the resistance to adopting such a technology. Park, Park [123] confirmed that using this type of application could have a positive impact on cost and sometimes increase the collaboration level. Another factor will be the security concern, as people will not feel secure in providing information about themselves, especially in conservative countries. Also, cargo owners will not easily trust unknown persons with their precious cargo. These results are aligned with Yiu, Grant [131], Laukkanen, Sinkkonen [132], and Li, Gong [133]. Furthermore, in third-world countries, last-mile is not taken from a business perspective as there is always an expected error from users, which highlights the complexity factors. Therefore, participants who perceived last-mile application as intricate and challenging to comprehend exhibited greater levels of resistance, which is consistent with Yiu, Grant [131], and Olsson, Hellström [21]. The last factor is scalability, in which the expected number of participants will not be large, which will lead to less availability of operating units and will reduce the delivery time of parcel owner's, which is always very important to them. This aligns with the proposed hypothesis that the resistance of given societies to last-mile delivery will increase when the CI, SC, C, and S increase [134].

## 10. Conclusions

This research aims to measure the problems faced when adopting last-mile applications in Alexandria, Egypt. It highlights the different barriers and their impact on the intention to adopt last-mile delivery. The research findings indicate that the negative impact of the different barriers was significant.

## 11. Theoretical Implications

From a methodological perspective, the research supports a unique contribution to knowledge by creating a model of a comprehensive approach to illustrating the factors that would affect adopting such a technology in a developing country. This study contributes to the literature on last-mile delivery adaptation by evaluating the barriers to adaptation in Alexandria, Egypt, a major city in a developing African country. The research tests different environmental, societal, and technological barriers to develop the framework and extend the use of TAM. TAM has been adopted and raised from many perspectives by various researchers, such as Arias-Oliva, Pelegrín-Borondo [135], and Tan and Sundarakani [136] (blockchain technology in the management), Agustina, Suprianto [137], Pazvant and Emel [138], and Liu, Li [139] (behavioural intentions regarding internet of things in adopting new devices and technologies), and Verma, Bhattacharyya [140], and Soon, Lee [141] (big data usage).

From a theoretical perspective, this research extends the TAM by testing the research variables in different contexts and environments by adopting Environmental, Societal, and Technological Barriers, and examining their impact on the intention to adopt last-mile logistics applications. Since last-mile delivery applications are not well explored in developing economies due to the lack of resources and knowledge [31], exploring the impact of these different barriers on the intention to adopt will help extend the theories, applying a deep analysis not explored in previous studies. This study also addresses a gap in the existing literature by adopting a comprehensive approach, categorising the barriers into technological, societal, and environmental factors. Furthermore, this study contributes to the comprehension of the relationship between these obstacles and their influence on the implementation of new technologies in the context of last-mile delivery. It highlights the necessity for additional research to examine the implications of these barriers on the societal acceptance and adoption of emerging technologies, particularly in developing nations.

From a policy maker perspective, this study has identified the key factors that can assist policy makers in making informed decisions by presenting the obstacles associated

with the implementation of last-mile practices. Furthermore, it offers guidance on the process of adopting a new technology and assists policy makers in formulating legislation that will facilitate the effective implementation of the application. Finally, they are expected to exercise heightened caution when faced with unfamiliar situations, particularly when dealing with complex last-mile mechanisms.

## 12. Practical Implications

The suggested model represents a new strategy for studying the capability of developing countries to adopt new technologies in response to the need to reduce environmental pollution and deterioration. As indicated in Ali, Gruchmann [24], and Ali, Melkonyan [25], developing countries need more sustainability practices to reduce pollution and environmental deterioration. Therefore, knowing the barriers that prevent the application a new technology would be a starting point for finding the right solutions.

The research findings demonstrated technological, societal, and environmental barriers' negative impact on society's resistance to accepting last-mile delivery. This will help the stakeholders reduce the negative effects of these findings to enhance the adoption of the last-mile delivery concept.

Many scholars concentrated on the adoption and how to design a different framework to adopt last-mile [66]. This means that last-mile stakeholders in Egypt still do not have enough knowledge and information related to their business to achieve satisfactory performance with the existing barriers. On the other hand, the study can help policy makers and practitioners integrate the gained knowledge and information to enhance the adoption processes of last-mile in developing North African countries. Finally, the study emphasised the importance of obtaining information about the barriers preventing societies' adoption of last-mile, sharing the information with all stakeholders and the influential role of this information in supporting this adoption.

## 13. Limitations and Future Research

This study only considers a specific geographical context (Egypt); therefore, future research must consider applying this study in different locations, as generalisability will be limited to countries with similar characteristics. However, it is imperative to take into account cultural differences. Whilst this study fills the gap mentioned by Marrucci et al. [142], more studies are needed to investigate the barriers affecting the intention to adopt last-mile delivery. Future research studies need to consider the previously mentioned barriers separately (environmental, societal, and technology) as the study used the impact of all these variables in general. Future research can identify mediating or moderating variables that could influence last-mile adoption, especially in developing economies. This study used a cross-sectional design that provides a snapshot of the relationships among the research variables. Future research can conduct a longitudinal study to explore the potential causality among variables. This study uses a snowball technique to collect data, which may cause sampling biases and systematic errors.

**Author Contributions:** Conceptualisation, M.A.S.; methodology, M.A.S. and A.H.A.; validation M.A.S.; formal analysis, M.A.S. and M.B.; investigation, M.A.S. and A.H.A.; resources, M.A.S. and T.K.; writing—original draft preparation, M.A.S.; writing—review and editing, M.A.S. and A.H.A.; supervision, A.H.A. and T.K. All authors have read and agreed to the published version of the manuscript.

**Funding:** This research received no external funding.

**Institutional Review Board Statement:** Not applicable.

**Informed Consent Statement:** Not applicable.

**Data Availability Statement:** The data supporting this study's findings are available on request from the corresponding author, M.A.S.

**Conflicts of Interest:** The authors declare no conflict of interest.

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
