# Peer review of "Barriers to Applying Last-Mile Logistics in the Egyptian Market: An Extension of the Technology Acceptance Model"

_sustainability, doi:10.3390/su151712748_

Round 1

Reviewer 1 Report

SUMMARY

Building on Technology Acceptance Model (TAM), the authors investigate the impact of social, environmental, and technological barriers on the adoption of the last-mile logistics application in the Egyptian market. Based on a questionnaire of 1060 entries, they found out that some elements could obstruct the implementation of last-mile delivery technologies: (1) complexity; (2) collaboration efforts between users and application developers; (3) technical knowledge and expertise impact on potential involved users. The results are useful for enhancing organizations in Egypt by using last-mile logistics as a tool for reducing contaminants, emissions, and carbon footprints. Furthermore, the authors introduce a tool that enhances customer satisfaction and reduces emissions by illustrating how last-mile logistics can be implemented, as a response to common implementation barriers in developing countries.

MAJOR COMMENTS

My main critiques are about the techniques used by the authors to collect data and the data itself. Is there bias in the collected/sampled data? Are there errors, missing values, or noise in it?

These are critical questions for evaluating the plausibility of the sampling methods used, which authors did not investigate and left for future work. However, most of the conclusions are highly dependent on the collected data.

In other words, if the data is biased or contains errors then these are inherited when drawing conclusions. As a result, it might happen that most conclusions are imprecise. The readers cannot verify this, also because there is no link to the data.

What is the data, what features does it have, and what questions were included in the questionnaire? The authors give several references (e.g. 78-83) related to these questions, but these references are for other applications and not last-mile logistics.

How does the data for adopting last-mile logistics differ from say the data for adopting autonomous vehicles? It would be nice to have a small sample of the data inserted in the paper.

MINOR COMMENTS

1.     Figure 1 is not placed correctly. Please fix its position on a single page.

2.     Why do the captions of Tables 1-3 and Figure 1 state only “This research”? They seem incomplete. Please describe what the tables and figures contain.

3.     Tables have captions at their bottoms and their tops. Please use one caption per table and be consistent in placing captions across tables.

4.     Tables are not aligned with the text. Please justify the table boundaries so that they fit the text body boundaries. Or, is there a reason for it?

5.     References are not aligned with the text but they take the entire page width. Is there a reason for it?

6.     The caption of Figure 2 should contain the meaning of each abbreviation used in the figure, i.e. What does RRL, RGA, TKA, etc. mean? This is regardless of the fact that the authors explain these abbreviations in the text earlier.

DECISION

The paper requires solid justification for the chosen data methods. The paper should also include more details about the questionnaire and the collected data. It is not clear whether there are biases in the data that may influence the conclusions of the paper. From this perspective, it is impossible to evaluate the significance of the research. The paper contains a number of minor textual issues, which prevent the smooth readability of the paper. Not least, the authors argue that their approach could be used in other cities with similar parameters, but they do not give examples of such cities, which prevents the reader from evaluating the generalizability of the proposed methods. Finally, I encourage the authors to polish their work much more before re-submitting it to this or any other journal.

There are minor issues that could be improved.

Reviewer 2 Report

The Article investigates the impact of social, environmental and technological barriers on adoption of the last-mile logistics application. The topic is very interesting.

Here are some suggestions:

1. Authors must do the proofreading of the Manuscript.

2. There are two type of citations. Must be only one.

3. At table 1 write the authors of the scales.

4. At table 2 correct the factor loadings.

5. Correct the References.

Authors must do the proofreading of the Manuscript.

Reviewer 3 Report

The article is good but needs substantial improvement. Please find comments.

1.       Provide motivation for the study in the introduction section. That is missing currently.

2.       There are a few typos and grammatical errors fix the same. The literature review needs to be updated in terms of sustainability aspects. What are different dimensions etc? Here are a few suggestions in this regard. “Green Lean Six Sigma for sustainability improvement: a systematic review and future research agenda "Integrating Green Lean Six Sigma and industry 4.0: a conceptual framework.

3.       Methodology section needs to be updated and needs a clear representation of the idea. Why authors have selected the present method and why not others?

4.       Results must be clear and discussed in detail related to the implications matters. How your results are comparable to the previous studies of the same nature must be presented.

5.       Implications need to be induced in terms of policymakers, practitioners and researchers

6.       Conclusion section needs to be revised in terms of the after-effects of the study.

fine

Round 2

Reviewer 3 Report

The authors have addressed all previous comments. Article ready for publication